# Learning to Guide Human Decision Makers with Vision-Language Models

## Abstract

There is increasing interest in developing AIs for assisting human decision making in *high-stakes* tasks, such as medical diagnosis, for the purpose of improving decision quality and reducing cognitive strain. Mainstream approaches team up an expert with a machine learning model to which safer decisions are offloaded, thus letting the former focus on cases that demand their attention. This **separation of responsibilities** setup, however, is inadequate for high-stakes scenarios. On the one hand, the expert may end up over-relying on the machine's decisions due to *anchoring bias*, thus losing the human oversight that is increasingly being required by regulatory agencies to ensure trustworthy AI. On the other hand, the expert is left entirely unassisted on the (typically hardest) decisions on which the model abstained. As a remedy, we introduce **learning to guide** (LTG), an alternative framework in which – rather than taking control from the human expert – the machine provides *guidance* useful for decision making, and the human is entirely responsible for coming up with a decision. In order to ensure guidance is *interpretable* and *task-specific*, we develop SLOG, an approach for turning *any* vision-language model into a capable generator of textual guidance by leveraging a modicum of human feedback. Our empirical evaluation highlights the promise of SLOG on a challenging, real-world medical diagnosis task.

## 1 Introduction

High-stakes applications in healthcare, criminal justice and policy making can substantially benefit from the introduction of AI technology, yet full automation in these scenarios is not desirable, due to ethical, safety and legal concerns, if not explicitly forbidden by law (Government of Canada, 2019; European Commission, 2021). For these reasons, human-AI or **Hybrid Decision Making** (HDM) is becoming increasingly popular to tackle high-stakes tasks. HDM algorithms pair a human decision maker with an AI agent – often a machine learning model – capable of providing support, with the goals of improving *decision quality* and lowering *cognitive effort*.

Most current approaches to HDM follow a principle of **separation of responsibilities**, in the sense that they route novel inputs to exactly one of the two agents – *either* the human *or* the AI – who is then responsible for coming up with a decision. Specifically, in existing approaches (Madras et al., 2018; Mozannar & Sontag, 2020; Keswani et al., 2022; Verma & Nalisnick, 2022; Liu et al., 2022; Wilder et al., 2021; De et al., 2020; Raghu et al., 2019; Okati et al., 2021), the AI first assesses whether an input can be handled in autonomy – e.g., it is low-risk or can be addressed with confidence – and defers to a human partner otherwise. These algorithms are beneficial in that they enable the human to focus on those cases that, according to the machine, most require their attention.

We argue that this setup is *suboptimal* and potentially *unsafe*. It is suboptimal because, whenever the machine opts for deferral, the human is left resolving hard cases completely unassisted (as in Fig. 1, right). At the same time, it is unsafe, because humans are affected by *anchoring bias* (Rastogi et al., 2022; Eigner & Händler, 2024), a phenomenon whereby decision makers tend to blindly rely on an initial impression (the anchor) and refrain from exploring alternative hypotheses. When the anchor is provided by an algorithm, the bias is amplified as humans tend to over-trust the machine's decisions when available and ignore their

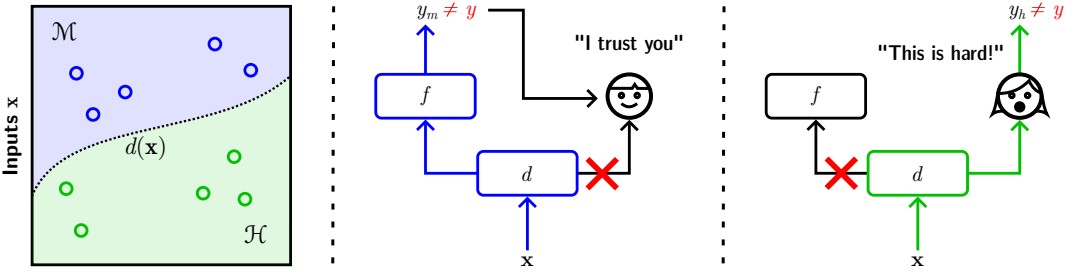

Figure 1: **Left**: Existing HDM algorithms employ a deferral function $d(\mathbf{x})$ to *partition* the input space $\mathcal{X}$ into $\mathcal{H}$ and $\mathcal{M}$. **Middle**: A predictor $f(\mathbf{x})$ handles the inputs falling in $\mathcal{M}$ (in **blue**). Because of *anchoring bias*, the human expert may end up blindly trusting its (possibly poor) decisions $y_m$. **Right**: The human, on the other hand, is left completely unassisted for those (possibly hard) decisions falling in $\mathcal{H}$, increasing the chance of mistakes in the human's decisions $y_h$ (in **green**).

own opinions, a phenomenon called *automation bias* (Cummings, 2012) (Fig. 1, middle). This effectively undermines *human oversight* over algorithmic decisions, which is increasingly being required by governments around the world to regulate the use of AI in high-stakes applications (Green, 2022).

As a remedy, we propose **learning to guide** (LTG), an alternative setup that side-steps these issues. In LTG the machine is trained to supply its human partner with interpretable *guidance* highlighting those aspects of the input that are useful for coming up with a high-quality decision. For instance, in pathology prediction, the guidance highlights the pathologies present in an input X-ray scan that are indicative of possible diagnoses. In LTG, *by construction*, all decisions are taken by the human expert – thus preventing automation bias – but facilitated by accompanying machine guidance.

We showcase LTG on **medical decision making** focusing on guidance formulated in **natural language**. To this end, we introduce SLOG (Surrogate-based Learning to Guide), an algorithm for turning large vision-language models (VLMs) (Radford et al., 2021; Yan & Pei, 2022; Sharma et al., 2021) into high-quality guidance generators. In a nutshell, SLOG takes a VLM pre-trained for caption generation and fine-tunes it using feedback about the quality of downstream human decisions inferred from generated guidance. SLOG keeps annotation costs under control by training a *surrogate model* that predicts downstream decision quality on a modest amount of feedback, and then using it to fine-tune the VLM in an end-to-end fashion. Our experiments on a challenging medical diagnosis task indicate that VLMs fine-tuned with SLOG output interpretable task-specific guidance that can be used to infer high-quality decisions.

**Contributions.** Summarizing, we:

- Identify serious flaws with existing HDM algorithms that compromise their applicability to high-stakes tasks.

- Introduce *learning to guide* (LTG), a novel approach for assisting human decision makers that ensures they are always in the loop.

- Develop SLOG, an LTG approach tailored for natural language guidance that can convert large VLMs into interpretable, task-specific guidance generators.

- Showcase the promise of SLOG on a challenging medical diagnosis task.

## 2 Hybrid Decision Making

We are concerned with decision tasks that, due to safety concerns, cannot be fully automated, such as medical diagnosis. Specifically, we focus on classification problems, with inputs $\mathbf{x} \in \mathbb{R}^d$ and categorical or multi-label decisions $y$.[1]

---

[1]While we focus on classification tasks, our remarks apply to other prediction problems, e.g., regression (De et al., 2020).

Research on Hybrid Decision Making (HDM) develops expert AI assistants capable of assisting human experts in such tasks. Considering the AI assistant and the human expert have different abilities, expertise, and biases, the central question of HDM is how to best integrate them.

**HDM by Separation of Responsibilities**. Existing HDM strategies solve this problem by following a principle of *separation of responsibilities*: any given instance $\mathbf{x}$ is assigned to exactly one of the two agents, who is then in charge of decision making, cf. Fig. 1. Specifically, they implement a *classifier* $f : \mathbf{x} \mapsto \hat{y}$, playing the role of an AI agent, as well as a *deferral policy* $d : \mathbb{R}^d \to \{\texttt{machine}, \texttt{human}\}$ that partitions the input domain $\mathcal{X}$ into two disjoint subsets, $\mathcal{M}$ and $\mathcal{H}$. Novel inputs $\mathbf{x}$ falling in the former are handled by $f$ and those falling in the latter are handled by the human expert. This setup is known under a variety of names, including *learning to defer* (Madras et al., 2018; Mozannar & Sontag, 2020), *learning under algorithmic triage* (Raghu et al., 2019; Okati et al., 2021), *learning under human assistance* (De et al., 2020; 2021), and *learning to complement* (Wilder et al., 2021; Bansal et al., 2021).

Approaches differ in how they partition the input space $\mathcal{X}$. Earlier methods build on *prediction with a reject option* (Cortes et al., 2016), in which the deferral policy $d$ observes all incoming instances $\mathbf{x}$ and offloads those about which the predictor $f$ is unsure (based on, e.g., predictive variance) (Raghu et al., 2019). Since $f$ is fixed, the partition is static and depends only on the self-assessed uncertainty of the predictor. Assuming the latter is sufficiently well calibrated (Kendall & Gal, 2017), this strategy can perform well in practice (Liu et al., 2022). The main drawback with this setup is that the partitioning accounts for the machine's performance only, neglecting the human's expertise and biases. Madras et al. (2018) improve on this by *learning* the deferral policy $d$ so that it optimizes some decision theoretic measure of *joint team performance*, thus explicitly taking the quality of human decisions into account. Follow-up works (De et al., 2020; 2021; Wilder et al., 2021) go one step further and train the deferral policy $d$ and the predictor $f$ *jointly*, so as to adapt one to the other.

This setup has been extended to incremental (Keswani et al., 2021) and sequential (Joshi et al., 2021) decision making, and to bandit feedback (Gao et al., 2021). Theoretical studies have analyzed the consistency (Mozannar & Sontag, 2020) and calibration (Verma & Nalisnick, 2022) of the HDM pipeline and the structure of optimal deferral policies (Okati et al., 2021).

**Issues with Separation of Responsibilities.** At a high level, the desiderata that an HDM strategy ought to satisfy are the following:

**D1**. *Complementarity.* It should leverage the complementary capabilities of each agent to obtain better decisions *on average*, or equally good decisions at a lower cognitive cost, than each agent individually.

**D2**. *Synergy.* It should combine the contributions of each agent to obtain better *individual* decisions, or equally good decisions at a lower cognitive cost, than each agent individually.

**D3**. *Reliability.* It should produce decisions that are more reliable than those made by each agent individually.

Existing HDM approaches aim specifically at enabling complementarity (**D1**). In fact, the main benefit of offloading decisions to an AI is that of lowering the human's cognitive effort. Moreover, depending on the relative performance of the predictor on inputs in $\mathcal{M}$ compared to the expert, they can also improve the quality of the team's decisions (on average across inputs, not necessarily for all inputs). Under suitable conditions, learning to defer can *provably* do so (Donahue et al., 2022).

However, approaches complying with separation of responsibilities completely overlook synergy (**D2**). When the machine outputs a decision, the human is tempted to simply stick to it, thus *over-relying on the machine's decisions*, because of the previously mentioned anchoring (Rastogi et al., 2022) and automation (Cummings, 2012) biases. Conversely, whenever the machine opts for deferral, *the human is left resolving hard cases completely unassisted*. This also compromises reliability (**D3**), which is key in high-stakes applications, thus hindering the applicability of HDM.

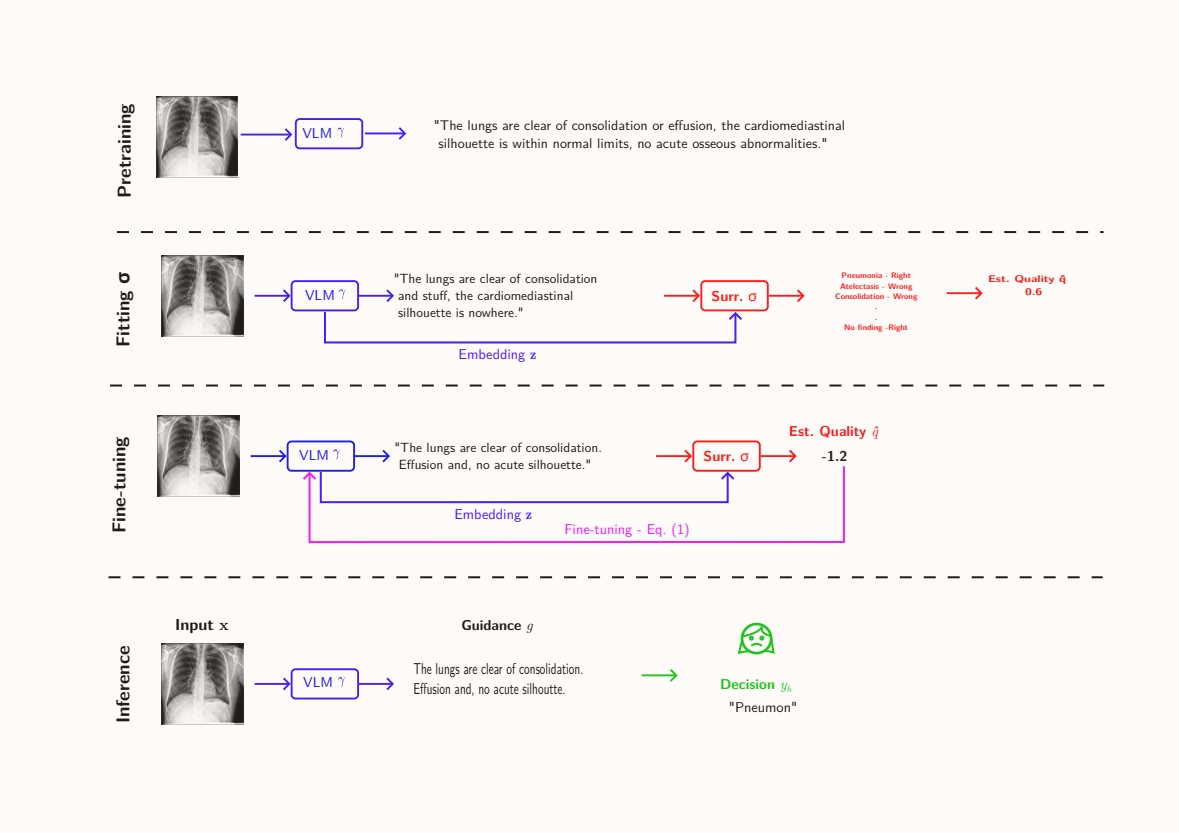

Figure 2: **The slog approach to learning to guide**. **Tier 1**: First we train a vlm to generate $g$ given an input **x**. **Tier 2**: The surrogate $\sigma_{\text{quality}}$ estimates the quality of the downstream decisions and it is trained using a modicum on annotated guidance-quality pairs. The $\sigma_{\text{quality}}$ is takes both the image and text as its input. **Tier 3**: Given a trained surrogate $\sigma_{\text{quality}}$, SLOG fine-tunes the VLM to output guidance $g$ achieving high (estimated) decision quality. **Tier 4**: Once the finetuning is done, SLOG can readily be used for inference.

## 3 Beyond Separation of Responsibilities with Learning to Guide

We propose *learning to guide* (LTG), a novel HDM framework that addresses the shortcomings of existing strategies by foregoing separation of responsibilities. In a nutshell, LTG aims to learn a *guidance generator* $\gamma(\mathbf{x})$, implemented as a machine learning model, that given an input **x**, outputs guidance $g$ that is useful for *assisting human decision making* on that input. In medical diagnosis, for instance, given a chest X-ray scan **x**, the guidance $g$ might describe pathology visible in the image that are useful for identifying pathologies and prescribe treatment, as shown in Fig. 2 (top). Critically, and in stark contrast with existing HDM approaches, in LTG the machine does not replace the human: *the final decision is always taken by the human partner, in collaboration with the AI*. This means the decision maker is always in the loop and responsible for the final decision.

**Desiderata for Guidance**. In order to support human decision making, guidance should satisfy the following natural properties:

**D4**. *Interpretability*. It should be *understandable* for the human expert at hand.

**D5**. *Informativeness*. It should be *informative* for the decision at hand.

If these are satisfied, then guidance can be used by human experts to address a specific downstream decision making task. Note that, satisfying these desiderata encourages satisfaction of **D1**–**D3**. In fact, if guidance is

interpretable (**D4**) and extracts decision-relevant elements from the input (**D5**), it should help the human in taking accurate decisions on individual instances (**D2**) and as a consequence improving the average quality of the decisions being made (**D1**). Additionally, interpretability helps the human in judging the quality of the guidance received, and thus evaluate the reliability of the overall decision (**D3**).

**Learning to Guide for VLMs**. In this paper we focus on *textual guidance* expressed in natural language as a mean to enable interpretability (**D4**). Motivated by their state-of-the-art performance in text generation tasks (Wei et al., 2022) and by their promise in pathological report generation (Shamshad et al., 2023; Chen et al., 2020; 2021; Hou et al., 2021; Kayser et al., 2022; Yunxiang et al., 2023; Bazi et al., 2023; Drozdov et al., 2020; Yan & Pei, 2022), we propose to leverage *vision-language models* (VLM) to generate guidance.

Off-the-shelf VLMs are not conceived for generating guidance for *specific* decision making tasks, and thus violate informativeness (**D5**). Clearly, a perfectly accurate medical report is also an optimal guidance for follow-up decisions, but generating highly accurate reports requires massive amounts of supervision, and reports generated by specialized VLMs are far from perfect (Shamshad et al., 2023).

The question is then how to *convert* such models into high-quality guidance generators. Focusing on (medical) decision making from image data, we address this problem by introducing SLOG, a novel approach for *turning vision-language models into guidance generators using human feedback* designed to comply with **D1**–**D5**. The rationale behind SLOG is to encourage VLMs to focus on accurately reporting those aspects of the input image that are most relevant *for the follow-up decisions*, possibly overlooking less important details. Next, we briefly discuss how SLOG uses annotations and then proceed to outline the main algorithm.

### 3.1 Estimating Downstream Decision Quality

Optimizing guidance for synergy (**D2**) requires knowing the quality of downstream decisions taken by a human expert supplied with the guidance itself. SLOG assumes access to quality ratings $\mathbf{q} \in [0, 1]^d$, where each $q_i$ encodes the quality of a downstream decision. For instance, if the expert has to determine the state of two conditions (e.g., "`pneumonia`" and "`fracture`"), then $d = 2$. Quality ratings for expert decisions can be obtained by comparing these against a gold standard (using, e.g., decision accuracy) or by consulting a second expert (using, e.g., a star rating system).

Clearly, there is a tension between the number of annotations necessary for fine-tuning a VLM and the cost of eliciting such annotations. SLOG addresses this issue by training a *surrogate model* $\sigma_{\text{quality}} : (\mathbf{x}, \mathbf{z}) \mapsto \widehat{\mathbf{q}}$ using a modicum of annotated quality ratings, and using it to estimate the quality of guidance $g$ generated by the VLM during fine-tuning. In practice, SLOG fits the surrogate on a training set $\mathcal{D}_{\text{surr}} = \{(\mathbf{x}_i, \mathbf{z}_i, q_i)\}$, where $\mathbf{x}_i$ is an input image, $\mathbf{z}_i$ is the embeddings of the VLM's guidance $g_i$ for that input, and $q_i$ is the quality of that guidance, by minimizing an average cross-entropy loss of the form:

$$\frac{1}{|\mathcal{D}_{\text{surr}}|} \sum_{(\mathbf{x}, \mathbf{z}, \mathbf{q}) \in \mathcal{D}_{\text{surr}}} \frac{1}{d} \sum_{i=1}^{d} \mathsf{CE}(q_i, \sigma_{\text{quality}}(\mathbf{x}, \mathbf{z})_i) \tag{1}$$

### 3.2 The slog Loop

In essence, SLOG takes a pre-trained VLM caption generator $\gamma$ and fine-tunes it for a number of rounds $T$. Let $\mathcal{D}_{\text{train}}$ be a data set of image-caption pairs (for instance, a subset of the data that $\gamma$ was trained on) and $\mathcal{D}_{\text{tune}}$ a larger set of unlabeled images from the target decision making task. In each round $t = 1, \ldots, T$, SLOG samples a batch $\{\mathbf{x}_1, \ldots, \mathbf{x}_B\}$ from $\mathcal{D}_{\text{tune}}$ uniformly at random with replacement, and computes guidance $g_i^t = \gamma(\mathbf{x}_i)$ and embeddings $\mathbf{z}_i^t$ for each input. Then, it evaluates the quality of the generated guidance using the frozen surrogate $\sigma_{\text{quality}}$ and fine-tunes the VLM $\gamma$ by minimizing an augmented loss of the form:

$$\mathsf{CE}(\gamma, \mathcal{D}_{\text{train}}) - \frac{\lambda}{|\mathcal{D}_{\text{tune}}|} \sum_{(\mathbf{x}, \mathbf{z}) \in \mathcal{D}_{\text{tune}}} \frac{1}{d} \sum_{i=1}^{d} \sigma_{\text{quality}}(\mathbf{x}, \mathbf{z})_i \tag{2}$$

for a given number of epochs. Eq. (2) trades off text generation performance on the training set $\mathcal{D}_{\text{train}}$ – so as to discourage catastrophic forgetting – and estimated quality of downstream decisions on the fine-tuning

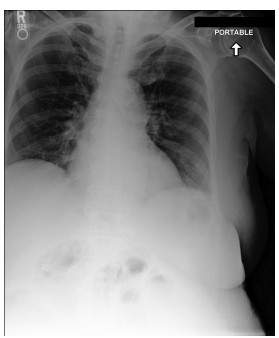

**Findings.** The right costophrenic angle is not imaged. Otherwise, the lungs are clear. The heart size is upper limits of normal. Enteric tube courses below the level of the diaphragm. There is no pneumothorax.

**Impression.** An enteric tube courses below the level of the diaphragm.

Figure 3: An example of a radiography along with its corresponding medical report consisting of *'findings'* and *'impression'*.

set $\mathcal{D}_{\text{tune}}$. Here, $\lambda > 0$ is a hyper-parameter. Fine-tuning then amounts to applying gradient descent to batches comprising training and fine-tuning examples in equal proportions. Once done, the SLOG loop then repeats. As long as the surrogate generalizes the quality rating annotations, the VLM gradually learns to output text that works well as guidance tailored for the target decision task.

### 3.3 Benefits and Limitations

In stark contrast with existing HDM strategies, SLOG ensures that the human receives guidance useful for decision making while keeping them in the loop. The cognitive load of LTG is entirely devoted to ensuring it can be safely employed in high-stakes applications, where there is little room for mistakes and humans *have* to be in control at all times (Zhang et al., 2020), as increasingly prescribed by legal frameworks (Government of Canada, 2019; European Commission, 2021). LTG and SLOG are designed explicitly for supporting HDM in these cases. SLOG is reminiscent of mainstream approaches to LLM alignment, such as reinforcement learning with human feedback (RLHF) (Ziegler et al., 2020; Ouyang et al., 2022), but differs from them in aims and technology. While RLHF strives to improve factuality and reduce harmfulness of generated content (Ouyang et al., 2022), ignoring the decisions these impinge on, SLOG specifically aims at improving quality of downstream human decisions for a specific decision making task. At the same time, SLOG foregoes reinforcement learning approaches (Schulman et al., 2017) in favor of a simpler and more direct end-to-end fine-tuning strategy.

One limitation of SLOG is that the performance of the guidance generator hinges on that of the surrogate, which in turn relies on the amount of quality ratings available for training. In Section 4 we present an ablation study showing that a limited amount of quality annotations are sufficient for SLOG to improve generated guidance. Another limitation of SLOG is that it currently assumes quality ratings are readily available, which is not always the case. One option is then to integrate SLOG with active learning strategies (Settles, 2012; Herde et al., 2021) to acquire informative quality ratings whenever needed. Doing so is however outside the scope of this paper and left to future work. Finally, the guidance output by VLMs may suffer from hallucination, that is, it may contain untrue statements. However, SLOG directly maximizes factuality of guidance on the fine-tuning set $\mathcal{D}_{\text{tune}}$, meaning that a simple way of reducing the chance of non-factual statements is to employ a larger fine-tuning set. This is relatively cheap to do, as no annotations are required. Moreover, large language models can be surprisingly well-calibrated (Kadavath et al., 2022), meaning that generated guidance can be filtered based on the VLM's own uncertainty estimates to prevent over-reliance (Eigner & Händler, 2024) and avoid low-quality decisions (Zhang et al., 2020).

## 4 Empirical Analysis

In this section, we answer empirically the following research questions:

**Q1**. Does SLOG improve the informativeness of generated guidance despite relying on a surrogate model?

Table 1: Comparison between the splits of the original data and our filtered data.

| | Original split | | | Our split | | |
|---|---|---|---|---|---|---|
| Components | Train | Val | Test | Train | Val | Test |
| Reports | 222,758 | 1,808 | 3,269 | 125,417 | 991 | 1,624 |
| Images | 368,960 | 2,991 | 5,159 | 232,855 | 1,837 | 2,872 |

**Q2**. Does SLOG improve the quality of the decisions being made using its guidance?

**Data set.** We evaluate SLOG on the `Mimic-CXR-IV` data set (Johnson et al., 2019), one of the largest publicly available medical decision data sets, consisting of $227,835$ radiology reports and $377,110$ chest X-ray scans. We focus on the *findings* and the *impression* sections of the reports. As shown in Fig. 3, the findings are text-based descriptions of what can be observed in the scan, and constitute the basis on top of which the expert forms their impression, i.e., their initial opinion about the potential pathologies of a patient. We discarded examples where either findings or impressions were not available, resulting in the *training*, *validation* and *test* splits presented in Table 1.

**Vision-language Models.** We consider the `R2Gen` (Chen et al., 2020) and `R2GenCMN` (Chen et al., 2022) as our candidate vision-language models. While the former is a memory-driven transformer specifically designed for pathological report generation from chest X-ray images, the latter uses a cross modal network (CMN) in order to achieve better mapping between diverse modalities. The `R2Gen` architecture builds on the observation that similar radiographs may correspond to reports sharing similar patterns. To exploit this, it employs a pre-trained CNN model to extract patch features and encodes these into hidden states with an encoder. A decoder then maps the hidden states into words at each time point with the help of a relational memory and memory-driven conditional layer normalization. The relational memory component allows the transformer to store and repurpose shared patterns and thus generate more coherent reports (Chen et al., 2020). Chen et al. (2022) pushes the argues that existing literature shows that there is limited scope for a proper alignment of different modalities. Addressing this issue, the authors developed a cross modal network where the encoded features of an image is fed to the CMN module to obtain the memory representations. A similar operation is done for the text embeddings. Thus, the shared information of the text and visual features can be stored in the memory. In particular, the CMN module employs a matrix where each row of the matrix is allotted for cross-modal memory information for image and texts.

**The decision-making task.** Our study focuses on a critical step of the medical decision process: making the right diagnosis.[2] The task consists in diagnosing pathologies (see Table 5 for the list of 14 possible pathologies) from X-ray images. In our experiments, we pre-train the `R2Gen` and `R2GenCMN` VLMs to predict *findings* from images, and then fine-tune with SLOG to automatically generate higher-quality guidance for a human decision maker. Given that the *impression* is the opinion that the expert forms about potential pathologies visible in the image, *we fine-tune our VLM to produce textual guidance that – once interpreted by a human expert – leads to the same diagnosis entailed by the impression.*

**Simulating the human expert.** SLOG can do so provided it has access to quality ratings of downstream human decisions. We simulate human decisions using a machine learning model, denoted HumanProxy, for reproducibility. Specifically, HumanProxy takes a scan **x** and a corresponding VLM-generated report and diagnoses the 14 candidate pathologies using three classes: definitely present (*positive*), definitely absent (*negative*), and unclear (*ambiguous*). Following (Lovelace & Mortazavi, 2020a), we implement HumanProxy as a classification model that takes both reports and images as inputs and train it on ground-truth labels obtained by applying the `CheXpert` (Irvin et al., 2019) automated annotation tool to the ground-truth *impressions*. Please note that while the $\sigma_{\text{quality}}$ surrogate is an integral part of SLOG, HumanProxy is an experimental detail necessary for evaluation.

---

[2]While strictly speaking medical diagnosis is not a decision making task, an incorrect diagnosis inevitably leads to poor decisions.

In order to emulate a setting with sparse human supervision, we assume *ground-truth labels are available for 10% of the training data only*. This ground-truth dataset $\mathcal{D}_{\text{surr}}$ is used for training both the model simulating human decisions HUMANPROXY and the quality surrogate model $\sigma_{\text{quality}}$ estimating the quality of these decisions. We rely on a stratified sampling procedure to select $\mathcal{D}_{\text{surr}}$ so as to maintain a reasonable coverage of the different classes.

Overall, we proceed as follows. First, HUMANPROXY is trained on $\mathcal{D}_{\text{surr}}$ to output a diagnosis given ground-truth findings (as these are the only ones for which we know the corresponding human diagnosis). Once trained, we use HUMANPROXY to produce quality ratings by computing the correctness of its predictions over $\mathcal{D}_{\text{surr}}$, which will later on be used for training the surrogate $\sigma_{\text{quality}}$.

To avoid biasing the quality rating supervision by computing it on training instances, we run $k$-fold cross validation on $\mathcal{D}_{\text{surr}}$ and collect quality ratings from the $k$ validation folds. For each validation fold, we compute decisions using both VLM-generated guidance and ground-truth text, so as to provide examples of both predicted and ground-truth guidance to train the quality surrogate model.

**Quality surrogate model.** As explained in Section 3.1, the surrogate $\sigma_{\text{quality}}$ should estimate the quality of human decisions when fed with the VLM guidance. In our experiments, human decisions are proxied with the 14 labels output by HUMANPROXY. The surrogate is thus trained to predict the *correctness* of each of the 14 predictions made by HUMANPROXY obtained in the previous step. Just like HUMANPROXY, the surrogate $\sigma_{\text{quality}}$ used by SLOG is also implemented as an attention-augmented LSTM and trained to minimize the average cross entropy loss on the ground-truth dataset described in the previous paragraph. Albeit having different purposes, the HUMANPROXY and $\sigma_{\text{quality}}$ fundamentally consist of similar model architecture. The multimodal functionality of both the models are established with a visual-encoder module and a text encoder module, where the former is a ResNet 101 based model that extracts the features of the radiology images and the later is a transformer based module that extracts nuanced features of the reports. To this end, we concatenate the features obtained from the text extractor and image extractor before applying a fully connected layer onto the concatenation.

**Overall pipeline.** Summarizing, our experimental pipeline is laid out as follows. In a first step, a `R2Gen` VLM is pre-trained on the training split $\mathcal{D}_{\text{train}}$ to generate findings. The VLM is then applied to the decision ground-truth dataset $\mathcal{D}_{\text{surr}}$ (10% of the training split) and the generated guidance is fed to HUMANPROXY to obtain (simulated) human decisions and corresponding quality annotations. This information is then used to fit the quality surrogate model $\sigma_{\text{quality}}$. Finally, the VLM is fine-tuned *on the entire training split* according to Eq. (2) for 10 epochs and evaluated on the test data. The source code is provided in the supplementary material. Fig. 4 depicts a pictorial summary of the overall pipeline.

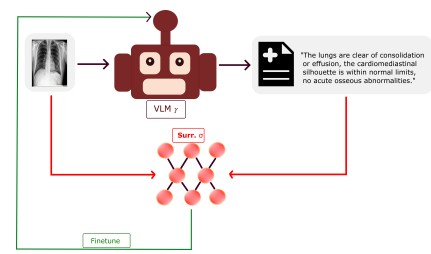

Figure 4: Finetuning policy of SLOG

## 4.1 Training the VLM

We trained our baseline VLM with an Nvidia A100 80GB GPU and with batch size 256. We restricted the maximum sequence length to 70 in order to avoid computational over head in later stages of our experiment. We fine-tuned the baseline models (`R2Gen` and `R2GenCMN`) on the same GPU with a batch size of 64. The values of all hyper-parameters were taken verbatim from (Chen et al., 2020) and (Chen et al., 2021), respectively. While training the baseline model, we used a patience of 20 and stored the best model with the highest BLEU-4 score.[3]

---

[3]The BLEU score is useful to evaluate the quality of machine-generated text with respect to a reference text, usually generated by a human. Specifically, BLEU-4 considers the overlap of each 4-gram between the machine-generated and the reference text (Papineni et al., 2002).

Table 2: **R2Gen: Outcomes from the test split used to evaluate** $\sigma_{\text{quality}}$. Results show per-class, macro and micro averaged precision, recall and $F_1$.

| PATHOLOGY | $Pr$ | $Rc$ | $F_1$ |
|---|---|---|---|
| No Findings | 54.88 | 87.53 | 67.46 |
| Cardiomediastinum | 84.68 | 93.47 | 88.85 |
| Cardiomegali | 90.24 | 94.11 | 92.13 |
| Lung Lesion | 78.28 | 93.07 | 85.03 |
| Lung Opacity | 92.07 | 94.21 | 93.13 |
| Edema | 92.52 | 93.87 | 93.19 |
| Consolidation | 91.99 | 92.71 | 92.35 |
| Pneumonia | 42.08 | 84.86 | 56.26 |
| Atelectasis | 82.87 | 94.67 | 88.38 |
| Pneumothorax | 60.89 | 91.55 | 73.13 |
| Pleural Effusion | 96.55 | 97.12 | 96.84 |
| Pleural Other | 62.86 | 87.97 | 73.33 |
| Fracture | 93.94 | 93.86 | 93.9 |
| Support Devices | 78.03 | 91.75 | 84.34 |
| MACRO | 92.96 | 80.84 | 86.19 |
| MICRO | 92.20 | 78.71 | 84.17 |

### 4.2 Surrogate training

In order to emulate a setting with sparse human supervision, we assume *ground-truth labels are available for 10% of the training data only.* This ground-truth dataset $\mathcal{D}_{\text{surr}}$ is used for training both the model simulating human decisions HUMANPROXY *and* the quality surrogate $\sigma_{\text{quality}}$ estimating the quality of these decisions. HUMANPROXY, which acts as a proxy for the human annotator, is a multimodal classification model which takes the radiology image and corresponding report as inputs and predicts the 14 target symptoms (cf. Table 5). To this end, in addition to calculating the loss and classification metrics, we conducted a sample-wise comparison between the ground truth labels and the prediction with the purpose of generating training data for $\sigma_{\text{quality}}$. This comparison yielded an $m \times n$ matrix $Y_q$ where $m = 14$ and $n$ is the number of training examples. Let us consider $G$ is the ground truth matrix of labels used for HUMANPROXY and $P$ is the matrix of labels predicted by the model on the validation data. We define,

$$Y_q = y_{ij}, \quad \text{where } y_{ij} = \begin{cases} 1 & \text{if } G_{ij}=P_{ij} \\ 0 & \text{Otherwise} \end{cases} \tag{3}$$

In order to train HUMANPROXY, we use k-fold cross validation with $k = 5$. We assessed the performance of our model on the validation set by scrutinizing the micro $F_1$ score pertaining to the positive mentions.

The $\sigma_{\text{quality}}$ takes same input as the HUMANPROXY, but instead of three labels, it outputs either 1 or 0 for the 14 classes (see Equation 3). In Table 2, we report the results obtained from the test split that was used to evaluate the performance of $\sigma_{\text{quality}}$.

### 4.3 Finetuning

We ran the finetuning for additional 10 epochs. In this phase, we froze both the $\sigma_{\text{quality}}$ and the visual encoder layer of the baseline R2Gen model. We tried with varying values of $\lambda$ and the best model was chosen based on the validation $F_1$ score. Eventually, for R2Gen, we chose 10 as the value of hyperparameter $\lambda$ as $\lambda = 10$ yielded the best $F_1$ during finetuning. Along with finetuning using $\lambda = 10$, we also experimented with $\lambda = 0$ to finetune without the $\sigma_{\text{quality}}$. In both cases, we finetuned the baseline model for equal number of epochs. We followed the exact same pipeline for R2GenCMN and chose 0.01 as the value for hyperparameter $\lambda$.

Table 3: `R2GenCMN`: Outcomes from the test split used to evaluate $\sigma_{\text{quality}}$ Results show per-class, macro and micro averaged precision, recall and $F_1$.

| PATHOLOGY | $Pr$ | $Rc$ | $F_1$ |
|---|---|---|---|
| No Finding | 86.22 | 79.69 | 82.82 |
| Cardiomediastinum | 92.65 | 93.24 | 92.94 |
| Cardiomegaly | 94.25 | 94.65 | 94.45 |
| Lung Lesion | 91.93 | 91.89 | 91.91 |
| Lung Opacity | 93.61 | 95.48 | 94.54 |
| Edema | 94.99 | 96.82 | 95.90 |
| Consolidation | 93.12 | 94.65 | 93.88 |
| Pneumonia | 74.99 | 75.51 | 75.25 |
| Atelectasis | 93.58 | 92.36 | 92.97 |
| Pneumothorax | 87.55 | 85.46 | 86.49 |
| Pleural Effusion | 96.73 | 98.38 | 97.55 |
| Pleural Other | 84.00 | 81.90 | 82.94 |
| Fracture | 93.62 | 94.36 | 93.99 |
| Support Devices | 91.46 | 88.19 | 89.80 |
| MACRO | 90.62 | 90.18 | 90.39 |
| MICRO | 91.18 | 90.84 | 91.01 |

Table 4: **slog boosts estimated quality of generated guidance without compromising text quality**. The results show that SLOG substantially improves estimated guidance quality as measured by the surrogate model ($\sigma_{\text{quality}}$) without affecting text quality as measured by Bleu scores over ground-truth caption data.

| MODEL | SETTING | BLEU$_1$ | BLEU$_2$ | BLEU$_3$ | BLEU$_4$ | BLEURT$_4$ | AVG LENGTH | $\sigma_{\text{quality}}$ |
|---|---|---|---|---|---|---|---|---|
| R2Gen | Pretrained | 0.36 | 0.22 | 0.15 | 0.11 | -0.38 | 45.93 | 0.39 |
| | Fine-tuned | 0.33 | 0.21 | 0.14 | 0.10 | -0.40 | 44.03 | 0.39 |
| | SLOG | 0.35 | 0.22 | 0.15 | 0.11 | -0.38 | 47.21 | 0.44 |
| R2GenCMN | Pretrained | 0.38 | 0.23 | 0.16 | 0.11 | -0.36 | 51.64 | 1.84 |
| | Fine-tuned | 0.37 | 0.22 | 0.15 | 0.11 | -0.35 | 50.22 | 1.85 |
| | SLOG | 0.38 | 0.23 | 0.16 | 0.11 | -0.34 | 49.70 | 2.02 |

## 4.4 Results

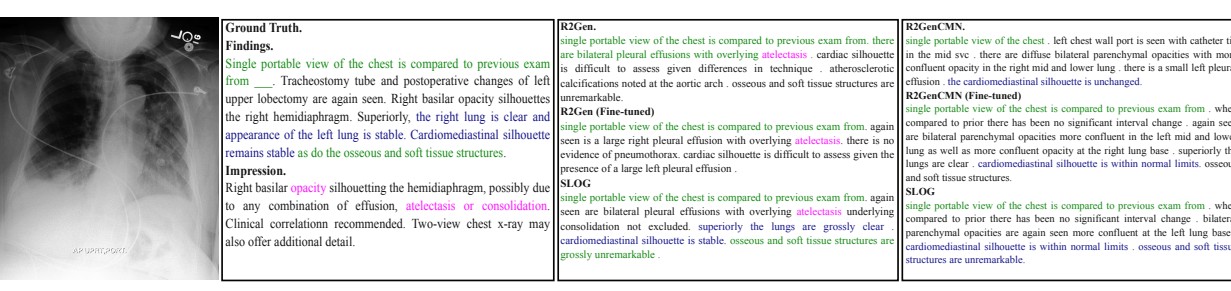

Figure 5: A qualitative example of the improvement of the guidance from SLOG with respect to the competitors. Green text indicates sentences that are (approximately) shared between the ground truth, SLOG and at least one of the competitors. Blue text indicates sentences shared between the ground truth findings (resp. impression) and SLOG, but missed by the competitors. No ground-truth sentences are shared between ground-truth text and competitors but missed by SLOG in this example.

**Q1: slog improves informativeness on the test set without compromising BLEU score.** A potential issue with using a surrogate model as a proxy of decision quality is that the fine-tuned VLM might end up overfitting the surrogate and produce guidance that, while seemingly informative, is unrelated to the actual input scan. SLOG prevents this by complementing estimated guidance quality as computed by the surrogate model with guidance appropriateness for the input image as measured by cross entropy over a training set of findings. Table 4 confirms the effectiveness of this strategy. SLOG substantially improves estimated guidance quality (second term in Eq. (2)) without compromising text quality, as measured both in terms of BLEU score and BLEURT (Sellam et al., 2020), over test examples. For the sake of fairness, we compared SLOG (with $\lambda = 10$) with both the pre-trained `R2Gen` model (before the fine-tuning stage), and the `R2Gen` model fine-tuned for the same number of epochs as SLOG, but with caption-level supervision only (i.e., setting $\lambda = 0$ in Eq. (2)), as well as with two `R2GenCMN` models fine-tuned in the same way. Fig. 5 shows a qualitative example of the improvement in guidance of SLOG with respect to the competitors. First, SLOG's guidance retains all pieces of text that any of the other approach shares with the ground-truth text (green text). On top of this, SLOG retrieves additional chunks of text that are shared with ground truth findings (blue text) and impression (magenta text), even if the latter are never explicitly included as training supervision, confirming the effectiveness of the quality surrogate in encouraging the generation of relevant guidance for the diagnosis.[4]

**Q2: SLOG improves quality of decisions.** Table 5 and Table 6 show the results in terms of decision quality, as measured by the $F_1$ score of the positive label for all 14 classes (multi-label prediction). Results clearly indicate the effectiveness of the SLOG guidance in improving decision quality, despite the little supervision it received. SLOG outperforms the competitors in 7 out of 14 classes. All methods fail to identify any positive occurrence for three particularly under-represented classes (`Pneumonia`, `Pleural Other`, `Fracture`), while SLOG slightly underperforms with respect to pre-trained `R2Gen` in 2 classes only. It is worth noticing that SLOG is especially effective in improving recall without affecting precision on average, as shown by the two bottom lines reporting results averaged over classes (macro) and instances (micro) respectively. Although `R2GenCMN` shows improvement in 5 out of 14 classes, SLOG under `R2GenCMN` also improves the quality of prediction in some of the rare classes (Lung lesion, Consolidation) . In short, SLOG applied to both `R2Gen` and `R2GenCMN` showcases an overall improvement in both micro and macro $F_1$.

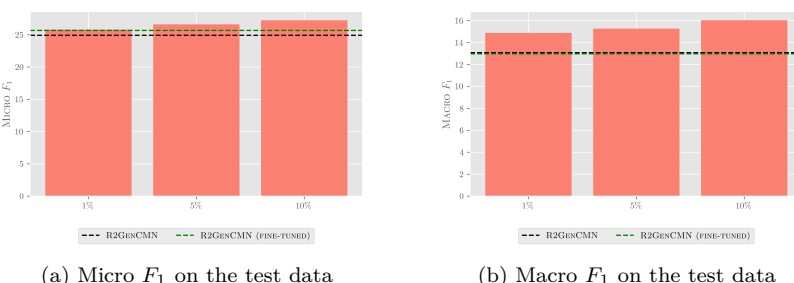

(a) Micro $F_1$ on the test data          (b) Macro $F_1$ on the test data

Figure 6: Performance of the finetuned models involving $\sigma_{\text{quality}}$ trained with varying percentage of train examples. The black and green lines are the $F_1$ scores obtained with the pretrained and finetuned model ($\lambda = 0$)

**Ablation.**

To investigate the influence of $\sigma_{\text{quality}}$ in the finetuning process, we conducted an ablation study based on the percentage of training data used to train HUMANPROXY and $\sigma_{\text{quality}}$. In our primary experiments, we took 10% of training examples to train the surrogate models. However, for the ablation, we finetuned the baseline model $\sigma_{\text{quality}}$ with only 1% and 5% of the training examples. The results can be viewed in Fig. 6. While the surrogates trained with 10% of the data understandably yield the best $F_1$, an improvement can still be observed when reducing the amount of training data further. In particular, macro $F_1$ (right) shows

---

[4]Notice that while the SLOG guidance text is longer than the one of the competitors in this example, with various chunks of text which are not obviously connected to ground-truth ones, its overall quality is still much higher.

Table 5: **SLOG boosts quality of downstream decisions.** Results show per-class, macro and micro averaged precision, recall and $F_1$. Best $F_1$ results are boldfaced. SLOG outperforms competitors on 7 out of 14 classes, is on-par in 5 (where no method manages to identify any positive occurrence) and worse on 2. On average, it outperforms competitors in both precision and recall (both micro and macro averaged).

| | R2Gen (pretrained) | | | R2Gen (fine-tuned) | | | SLOG | | |
|---|---|---|---|---|---|---|---|---|---|
| PATHOLOGY | $Pr$ | $Rc$ | $F_1$ | $Pr$ | $Rc$ | $F_1$ | $Pr$ | $Rc$ | $F_1$ |
| No Findings | 34.42 | 59.73 | 43.67 | 33.08 | 59.95 | 42.64 | 38.09 | 54.98 | **45** |
| Cardiomediastinum | 0 | 0 | 0 | 1.92 | 5.56 | **2.86** | 0 | 0 | 0 |
| Cardiomegaly | 14.06 | 28.72 | 18.88 | 14.29 | 23.94 | 17.89 | 15.7 | 37.23 | **22.08** |
| Lung Lesion | 0 | 0 | 0 | 0 | 0 | 0 | 0 | 0 | 0 |
| Lung Opacity | 26.32 | 11 | 15.52 | 29.63 | 9.78 | 14.71 | 30.52 | 18.58 | **23.1** |
| Edema | 42.03 | 18.65 | **25.84** | 40 | 16.08 | 22.94 | 43.48 | 16.08 | 23.47 |
| Consolidation | 0 | 0 | 0 | 3.23 | 1.41 | 1.96 | 9.43 | 7.04 | **8.06** |
| Pneumonia | 0 | 0 | 0 | 0 | 0 | 0 | 0 | 0 | 0 |
| Atelectasis | 17.65 | 22.57 | 19.81 | 17.5 | 18.58 | 18.03 | 19.88 | 28.76 | **23.51** |
| Pneumothorax | 0 | 0 | 0 | 8.33 | 3.33 | 4.76 | 13.64 | 10 | **11.54** |
| Pleural Effusion | 48.82 | 28.3 | **35.83** | 44.39 | 23.9 | 31.07 | 45.02 | 28.57 | 34.96 |
| Pleural Other | 0 | 0 | 0 | 0 | 0 | 0 | 0 | 0 | 0 |
| Fracture | 0 | 0 | 0 | 0 | 0 | 0 | 0 | 0 | 0 |
| Support Devices | 17.64 | 34.66 | 23.64 | 19.14 | 35.23 | 24.8 | 18.93 | 42.05 | **26.1** |
| MACRO | 14.37 | 14.55 | 13.08 | 15.11 | 14.13 | 12.98 | 16.76 | 17.38 | **15.56** |
| MICRO | 26.72 | 25.41 | 26.05 | 26.57 | 23.73 | 25.07 | 27.19 | 27.57 | **27.38** |

Table 6: Performance of R2GenCMN finetuned with SLOG. Results show per-class, macro and micro averaged precision, recall and $F_1$. Best $F_1$ results are boldfaced. SLOG outperforms competitors on 5 out of 14 classes, is on-par in 3 (where no method manages to identify any positive occurrence) and worse on 6. On average, it outperforms competitors in both precision and recall (both micro and macro averaged).

| | R2GenCMN | | | R2GenCMN (fine-tuned) | | | SLOG | | |
|---|---|---|---|---|---|---|---|---|---|
| PATHOLOGY | $Pr$ | $Rc$ | $F_1$ | $Pr$ | $Rc$ | $F_1$ | $Pr$ | $Rc$ | $F_1$ |
| No Findings | 33.93 | 43.21 | 38.01 | 36.03 | 48.42 | **41.31** | 36.11 | 47.06 | 40.86 |
| Cardiomediastinum | 1.92 | 5.56 | **2.86** | 0.0 | 0.0 | 0.0 | 1.27 | 5.56 | 2.06 |
| Cardiomegaly | 16.33 | 55.85 | 25.27 | 17.65 | 57.45 | **27.00** | 17.2 | 51.6 | 25.8 |
| Lung Lesion | 0.0 | 0.0 | 0.0 | 25.0 | 1.67 | 3.12 | 33.33 | 1.67 | **3.17** |
| Lung Opacity | 22.96 | 21.27 | 22.08 | 25.29 | 15.89 | 19.52 | 28.41 | 14.91 | **26.28** |
| Edema | 35.0 | 6.75 | 11.32 | 35.71 | 12.86 | 18.91 | 43.55 | 17.36 | **24.83** |
| Consolidation | 10.0 | 9.86 | 9.93 | 7.29 | 9.86 | 8.38 | 10.11 | 12.68 | **11.25** |
| Pneumonia | 28.57 | 1.31 | 2.5 | 19.23 | 3.27 | **5.59** | 14.29 | 1.96 | 3.45 |
| Atelectasis | 19.9 | 34.51 | **25.24** | 19.94 | 28.32 | 23.4 | 19.58 | 32.74 | 24.5 |
| Pneumothorax | 0.0 | 0.0 | 0.0 | 0.0 | 0.0 | 0.0 | 0.0 | 0.0 | 0.0 |
| Pleural Effusion | 50.29 | 24.18 | 32.65 | 44.39 | 25.0 | 31.99 | 48.55 | 29.95 | **37.01** |
| Pleural Other | 0.0 | 0.0 | 0.0 | 0.0 | 0.0 | 0.0 | 0.0 | 0.0 | 0.0 |
| Fracture | 0.0 | 0.0 | 0.0 | 0.0 | 0.0 | 0.0 | 0.0 | 0.0 | 0.0 |
| Support Devices | 17.95 | 48.86 | **26.26** | 18.18 | 43.18 | 25.59 | 17.61 | 44.32 | 25.2 |
| MACRO | 16.92 | 17.95 | 14.01 | 17.77 | 17.57 | 14.63 | 19.28 | 19.24 | **16.03** |
| MICRO | 23.44 | 26.61 | 24.93 | 24.64 | 26.81 | 25.68 | 25.56 | 29.32 | **27.31** |

that even with 1% surrogate training data, SLOG still manages to outperform both baselines (dashed lines), while micro $F_1$ (left) is on-par with them. This indicates that even the less informed surrogates can help

Table 7: Ablation study on dataset size for SLOG. Performance improves consistently as training data increases from 1% to 10%.

| Type | Metric | SLOG 1% | SLOG 5% | SLOG 10% |
|------|--------|---------|---------|----------|
| **Macro** | Precision | 19.11 | 15.65 | 19.28 |
| | Recall | 17.58 | 18.38 | 19.24 |
| | $F_1$ | 14.88 | 15.27 | **16.03** |
| **Micro** | Precision | 24.40 | 24.93 | 25.56 |
| | Recall | 27.57 | 28.57 | 29.32 |
| | $F_1$ | 25.89 | 26.62 | **27.41** |

improving prediction quality for the rarer – but possibly significant for decision making – classes. In Table 7, we notice that SLOG trained with 10% of training examples outperforms the former two in both micro and macro metrics. Detailed results can be obtained in Appendix A.

## 5  Related Work

**Aligning LLMs.** The standard approach for aligning large language models to human interests is reinforcement learning with human feedback (RLHF) (Ziegler et al., 2020; Ouyang et al., 2022). Several RLHF-based approaches that target medical tasks exist. Yunxiang et al. (2023) and Wang et al. (2023a) proposed medical chat models obtained by fine-tuning existing architectures, while Bazi et al. (2023) introduced a specially designed vision transformer. Seo et al. (2020) presented a method for improving the performance of an image caption generator with offline human feedback. SLOG can be viewed as a variant of RLHF that foregoes reinforcement learning in favor of a fully end-to-end fine-tuning strategy, for efficiency. It also differs in aim, in that it optimizes the model's guidance for *a specific human decision making task*, rather than for factuality and fairness in general (Ouyang et al., 2022).

**Pathological report generation.** Several approaches (Hou et al., 2021; Chen et al., 2020; Wang et al., 2022; 2023b) have been developed for machine-driven pathological report generation from chest X-ray images using the `Mimic-CXR-IV` (Johnson et al., 2019) and the Indiana University chest X-ray data sets (Demner-Fushman et al., 2016). (Lovelace & Mortazavi, 2020b) designed a model with a similar objective but they proposed to leverage the `CheXpert` dataset to enhance the coherence of their model. (Tanida et al., 2023) introduced a region-guided model to generate pathological reports, thus opening the window of interactive human-guided report generation. (Srivastav et al., 2024) used a large language model based on Vicuna-7B to generate radiology reports out of CXR images. A slightly different approach was used by (Woźnicki et al., 2024) where the authors used a large language model to extract the structured information out of the *findings* section of a report. However, these models are not concerned with optimizing the utility of the generated reports for the follow-up decision making. Our approach builds on top of these methods, enriching them with the ability to incorporate surrogate quality information (we use (Chen et al., 2020) in our evaluation, but any of these models can be adapted for SLOG).

**Other approaches.** LTG is related to *explain then predict* (ETP) (Camburu et al., 2018; Kumar & Talukdar, 2020; Zhao & Vydiswaran, 2021), a framework for building explainable (Guidotti et al., 2018) models in which a machine first outputs a full-fledged explanation – playing the role of "guidance" – and then derives a prediction from the explanation itself. In LTG, however, the prediction step is carried out by a human expert, and as such it is not differentiable. Also, ETP requires direct supervision on the explanations themselves, which is seldom available. In contrast, SLOG improves guidance quality using indirect scoring feedback, which is comparatively easier to acquire.

Finally, LTG is not restricted to textual guidance. One option is, for instance, to implement guidance in terms of explanations extracted from (or output by) an underlying image classifier to guide human decision making (Guidotti et al., 2018). From this perspective, LTG is tied to explanatory interactive learning (XIL) (Schramowski et al., 2020; Teso et al., 2023), in which the goal is to improve the quality of explanations

output by a machine learning model by interactively acquiring corrections to the explanations themselves. The key difference is that LTG focuses on down-stream decision quality and SLOG supports textual guidance, while XIL aims at more generally improving explanation quality and implementations do no support textual explanations.

## 6 Conclusion

We introduced *learning to guide* as an alternative setup for high-stakes hybrid decision making that ensures the human expert is always in the loop, as well as SLOG, an end-to-end approach for turning pre-trained VLMs into high-quality textual guidance using human feedback. Our results suggest that SLOG is effective at steering VLMs towards generating more informative guidance, leading to improved accuracy in downstream decisions.

In follow-up work, we plan to extend SLOG by integrating ideas from active learning (Settles, 2012) to acquire the quality rankings, and to explore connections with explainable AI (Guidotti et al., 2018), explanatory interactive learning (Schramowski et al., 2020; Teso et al., 2023) and skeptical learning (Zeni et al., 2019) to facilitate the identification and correction of potential issues with the generated guidance.

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

# A  Appendix

Table 8: Ablation study of SLOG based on `R2GenCMN` with different proportions of labeled data (SLOG 1%, 5%, and 10%).

| Pathology | SLOG 1% | | | SLOG 5% | | | SLOG 10% | | |
|---|---|---|---|---|---|---|---|---|---|
| | Pr | Rc | $F_1$ | Pr | Rc | $F_1$ | Pr | Rc | $F_1$ |
| No Finding | 35.21 | 45.25 | 39.60 | 37.07 | 46.38 | 41.21 | 36.11 | 47.06 | 40.86 |
| Cardiomediastinum | 0.00 | 0.00 | 0.00 | 0.00 | 0.00 | 0.00 | 1.27 | 5.56 | 2.06 |
| Cardiomegaly | 15.87 | 49.47 | 24.03 | 16.67 | 51.60 | 25.19 | 17.20 | 51.60 | 25.80 |
| Lung Lesion | 50.00 | 1.67 | 3.23 | 0.00 | 0.00 | 0.00 | 33.33 | 1.67 | 3.17 |
| Lung Opacity | 26.79 | 21.03 | 23.56 | 26.49 | 21.76 | 23.89 | 28.41 | 14.91 | 26.28 |
| Edema | 41.98 | 17.68 | 24.89 | 41.61 | 18.33 | 25.45 | 43.55 | 17.36 | 24.83 |
| Consolidation | 8.43 | 9.86 | 9.09 | 10.75 | 14.08 | 12.20 | 10.11 | 12.68 | 11.25 |
| Pneumonia | 7.14 | 0.65 | 1.20 | 4.17 | 0.65 | 1.13 | 14.29 | 1.96 | 3.45 |
| Atelectasis | 17.24 | 26.55 | 20.91 | 19.17 | 30.53 | 23.55 | 19.58 | 32.74 | 24.50 |
| Pneumothorax | 0.00 | 0.00 | 0.00 | 0.00 | 0.00 | 0.00 | 0.00 | 0.00 | 0.00 |
| Pleural Effusion | 47.41 | 30.22 | 36.91 | 46.03 | 30.22 | 36.48 | 48.55 | 29.95 | 37.01 |
| Pleural Other | 0.00 | 0.00 | 0.00 | 0.00 | 0.00 | 0.00 | 0.00 | 0.00 | 0.00 |
| Fracture | 0.00 | 0.00 | 0.00 | 0.00 | 0.00 | 0.00 | 0.00 | 0.00 | 0.00 |
| Support Devices | 17.38 | 43.75 | 24.88 | 17.15 | 43.75 | 24.64 | 17.61 | 44.32 | 25.20 |
| **Macro Avg** | 19.11 | 17.58 | 14.88 | 15.65 | 18.38 | 15.27 | 19.28 | 19.24 | 16.03 |
| **Micro Avg** | 24.40 | 27.57 | 25.89 | 24.93 | 28.57 | 26.62 | 25.56 | 29.32 | 27.41 |

