# OpenReview forum: "Learning to Guide Human Decision Makers with Vision-Language Models"
_TMLR — Rejected by TMLR_

### Review · Reviewer_byyh · 2025-03-11

**Summary Of Contributions:**

The paper introduces learning to guid (LTG), a human-in-the-loop framework that enables machine learning models to provide guidance useful for human decision-making in Hybrid Decision Making (HDM) scenarios. The paper identifies the existing flaws in existing HDM frameworks where humans might be biased towards models' decisions or the complementary knowledge of humans and models are not used. The paper proposes Surrogate-based Learning to Guide (SLOG), which fine-tunes a VLM to provide informative guidance for humans to make the final decisions. Experiment results show the promise of the proposed approaches.

**Audience:**

Yes

**Claims And Evidence:**

No

**Requested Changes:**

I think the topic is interesting, and the paper is well-motivated. However, I’m mainly concerned about the evaluation part of the paper. Can the author provide further justification for the following:

- **Human experts simulation**: To what extent can the machine learning model simulate real human experts? Do real-world human experts benefit from the guidance generated by VLMs? The author is suggested to provide evidence as to why these experiment setups can be considered as high fidelity or provide experiments to justify that real human expert can indeed achieve higher performance given the guidance generated by VLMs.
- **Quality model**: In Table 3, the performance boost is evaluated by the quality model score. However, given that the ground truth is available, can the author directly show the performance improvement on the ground truth, which is stronger and more straightforward evidence?
- **Evaluated model and tasks**. Can the author provide more experiments on more VLMs and even datasets to show the generality of the proposed approach?
- **Clarity issue**. I strongly suggest the author include a figure showing the overall pipeline of the training process, given that the current version is a little bit hard to follow. Additionally, for $D_{train}$ and $D_{tune}$, the author mentions $D_{train}$ is a dataset of image-caption pairs the original VLMs are trained on, and $D_{tune}$ represents the unlabeled target tasks. However, in Sec 4, the author mentions pre-training R2Gen to predict findings, which is confusing: Can the author clarify what the $D_{train}$ and $D_{tune}$ in the paper are, respectively? Is the VLMs pre-train dataset (ie, predict findings) the same as the fine-tune dataset?

**Strengths And Weaknesses:**

### Strengths

- Interesting topics. The paper discusses an interesting topic where ML models are providing guidance to human experts to make decisions instead of solely relying on either models or humans.
- Well motivated. In real-world high-stakes applications, there might be an ethics issue to solely use ML for decision-making; the proposed approach can always ensure human-in-the-loop while benefitting from informative guidance of ML models.

### Weakness

- Unjustified experimental setups. The paper uses a multi-model machine learning model (ie, a classifier) to simulate human experts. However, how well the ML model simulates human behavior is not experimentally justified, leaving the fidelity of the setups unknown.
- Lack of enough evidence. The paper shows the quality of generated guidance using the score given by the quality model. It’s suggested to directly use the ground truth as a metric.
- Lack of evaluated models and datasets. The paper mainly uses one VLM on one dataset.

### Minor

- Some typos. In the last paragraph of Sec. 4.2, it should be “In Table 2”? In the first paragraph of Sec. 4.3, \lambda is not correctly defined.
- Not easy to follow.

---

> ### Author Response · Authors · 2025-04-26
>
> > Human expert simulations: To what extent can the machine learning model simulate real human experts?
>
> We apologize for the confusion.  The SLOG algorithm relies on a single surrogate model, $\sigma_{quality}$, which is trained to mimic human quality judgements about generated guidance and used for fine-tuning the VLM to produce better guidance.
>
> This is not the same as the $\sigma_{human}$ surrogate, which predicts _diagnoses_ instead. In short,
>
> First, we train $\sigma_{human}$ to **predict diagnoses** and apply it to each input. We then compare the diagnoses it predicted and the ground-truth ones to obtain a dataset of triplets (input, predicted diagnosis, is this diagnosis correct?) triplets. The last element is a binary label.
> Second, we train $\sigma_{quality}$ to predict the “is this correct?” label. We do so using a *limited number* of triplets. So our claim that SLOG relies on reasonable amounts of human quality judgments holds true.
>
> Note that $\sigma_{human}$ is just an experimental detail: in practice, diagnoses would be collected from an actual human decision maker.   Moreover, it is perfectly fine if part of the predicted diagnoses are *incorrect*, as these serve as negative labels for informing training of $\sigma_{quality}$.
>
> In contrast, $\sigma_{quality}$ is an integral part of the SLOG fine-tuning step.
>
> We realize that our naming choices were less than ideal, and renamed $\sigma_{human}$ to HumanProxy to avoid further confusion.
>
> > Quality model: In Table 3, the performance boost is evaluated by the quality model score. However, given that the ground truth is available, can the author directly show the performance improvement on the ground truth, which is stronger and more straightforward evidence?
>
> Unfortunately, we cannot provide any evaluation on this particular case the the ‘gold standard guidance’ is actually absent. As this guidance is always absent (at least in the current set up), we approximate the entire pipeline with respect to the ground truth ‘findings’
>
> > Evaluated model and tasks. Can the author provide more experiments on more VLMs and even datasets to show the generality of the proposed approach?
>
> Model. We appreciate your feedback and evaluated the performance of our pipeline SLOG with another model named R2GenCMN. We have updated the paper with the results of R2GenCMN. We also updated the empirical output image with the captions generated with R2GenCMN. We updated the results of R2GenCMN in Table 3, Table 4 and Table 6.
>
> Dataset.As for other medical tasks, mimic is pretty big already and there are not many alternatives that we can use. However, we evaluated the pipeline with a different model and showed that the SLOG outperforms the corresponding baseline models.
>
> > Additional figure
>
> Good point.  We have included a comprehensive illustration of the overall SLOG pipeline in Section 3. and also in Section 4 (Figure 4)
>
> > $\cal D_{train}$ vs. $\cal D_{tune}$
>
> We agree this is confusing. In Section 3, we introduced SLOG in full generality: in general, it may be possible that the data used for training (which needs to be annotated with quality annotations) and the data used for fine-tuning (which can be unsupervised) can differ. However, this needs not be the case: the training data *can* be used for fine-tuning, as it contains all the necessary information. This does not compromise the fairness of the evaluation, it only means that we fine-tune the VLM using inputs from the training set.

---

### Review · Reviewer_nG23 · 2025-03-14

**Summary Of Contributions:**

The paper is about AI-assisted (human) decision making. The authors introduce "learning to guide (LTG)", a framework where the AI model provides guidance instead of predictions that are optimized to be useful for decision making. They propose SLOG, an approach for fine-tuning vision-language models for the purpose of learning to guide and empirically evaluate this approach on a medical diagnosis task.

**Audience:**

Yes

**Broader Impact Concerns:**

No broader impact concerns

**Claims And Evidence:**

Yes

**Requested Changes:**

- This sentence is unclear to me. "the surrogate is also implemented as an attention-augmented LSTM and trained to minimize the average cross entropy loss on the ground-truth dataset described in the previous paragraph."
I understand this means that you train the quality surrogate model with a cross entropy loss on top of the loss described in equation 1, is this correct? If yes, I find this somewhat misleading with what is stated in section 3.1. and 3.3. since not only a "limited amount of quality annotations" are used to train the quality surrogate model in the experiment but also some ground-truth guidance that might not be available in all use cases. If this is the case, this should be made more clear in sections 3.1. and 3.3.
- I think the BLEU score should be explained as the paper is not explicit only about LLMs
- In Section 4 there are multiple small mistakes/typos that should be fixed (e.g., Table 2 caption "... are boldfaced" $\to$ no numbers are bold, Table 4 caption $\to$ numbers don't sum up to 14 classes, class "Lung Lesion" also has 0 scores but not mentioned in the text, typos in main text of this section as well)
- wrong citation format used in section 5 (use "\citet" when the citation is part of the text)
- typo $z_I$ in section 3.1

**Strengths And Weaknesses:**

Strengths:
- the paper is mostly well written and clear
- The proposed framework of learning to guide is nice and the application in the experiments is a well-chosen example for its use case

Weaknesses:
- Since there is only a purely empirical evaluation of SLOG, but the authors frame as a more general approach, it would have been nice to see experiments on more than one application

---

> ### Author Response · Authors · 2025-04-26
>
> Thank you for taking the time to evaluate our contribution.  We are glad that you found it well motivated and significant.  Below, we briefly reply to your remarks.
>
> >“the surrogate is also implemented as an attention-augmented LSTM and trained to minimize the average cross entropy loss on the ground-truth dataset” [...] I find this somewhat misleading with what is stated in section 3.1. and 3.3. since not only a "limited amount of quality annotations"
>
> We apologize for the confusion.  The SLOG algorithm relies on a single surrogate model, $\sigma_{quality}$, which is trained to mimic human quality judgements about generated guidance and used for fine-tuning the VLM to produce better guidance.
>
> This is not the same as the $\sigma_{human}$ surrogate, which predicts _diagnoses_ instead. In short,
>
> First, we train $\sigma_{human}$ to **predict diagnoses** and apply it to each input. We then compare the diagnoses it predicted and the ground-truth ones to obtain a dataset of triplets (input, predicted diagnosis, is this diagnosis correct?) triplets. The last element is a binary label.
> Second, we train $\sigma_{quality}$ to predict the “is this correct?” label. We do so using a *limited number* of triplets. So our claim that SLOG relies on reasonable amounts of human quality judgments holds true.
>
> Note that $\sigma_{human}$ is just an experimental detail: in practice, diagnoses would be collected from an actual human decision maker.   Moreover, it is perfectly fine if part of the predicted diagnoses are *incorrect*, as these serve as negative labels for informing training of $\sigma_{quality}$.
>
> In contrast, $\sigma_{quality}$ is an integral part of the SLOG fine-tuning step.
>
> We realize that our naming choices were less than ideal, and renamed $\sigma_{human}$ to HumanProxy to avoid further confusion.
>
> >Missing definition of BLEU score
>
> Thank you for pointing this out.  We now introduce the BLEU score in Section 4.1, line 539.
>
> >Other typos
>
> We have thoroughly proofread the text, and especially Section 4, and addressed all the typos you pointed out including the citations and all the others we found.  Please let us know if you find any other issues.

---

### Review · Reviewer_3fxa · 2025-04-12

**Summary Of Contributions:**

The paper introduces the learning to guide (LTG) paradigm for human-AI collaboration in high-stakes decision-making domains, such as medical diagnosis. Unlike conventional hybrid decision-making (HDM) approaches where decisions are either fully automated or entirely deferred to humans, LTG always leaves the final decision in human hands, with the model offering task-specific, interpretable guidance for better decision-making. LTG mitigates risks associated with conventional HDM such as over-reliance on automation or lack of support in challenging cases.

The authors proposed SLOG (Surrogate-based Learning to Guide), a fine-tuning strategy for adapting vision-language models (VLMs) to function as effective guidance generators. SLOG involves training a surrogate model that predicts the quality of human decisions based on textual guidance from available ground-truth annotations. This surrogate is then used to fine-tune the VLM by jointly optimizing for both the informativeness (using cross-entropy loss) and relevance (using surrogate estimate quality) of the guidance.

The authors conduct experiments on the MIMIC-CXR-IV dataset consisting of radiology reports and chest X-ray scans with the R2Gen model. The results demonstrate that SLOG improves the informativeness of the textual guidance that is generated and thus in turn boosts the quality of downstream decisions.

**Audience:**

Yes

**Claims And Evidence:**

Yes

**Requested Changes:**

See the weaknesses above for the list of proposed changes.

The authors must fully address weaknesses W2, W3 and W4 for me to recommend this work for acceptance. Additionally, the analysis of the generated guidance should be strengthened using automated techniques, as outlined in W5 and W6.

While I understand that the human evaluation suggested in W5 can be costly and potentially infeasible, conducting a small-scale human evaluation -- covering a few hundred images -- would be a valuable effort.

**Strengths And Weaknesses:**

**Strengths**

S1. The authors propose an alternative paradigm for human-AI collaboration offering a third way between full automation and complete delegation. This approach retains human control over decision-making while addressing critical ethical, safety, and legal concerns, particularly in high-stakes domains, and making it more trustworthy.

S2. SLOG focuses on generating task-specific guidance that is both actionable and optimized for effective decision-making. This capability is broadly valuable but especially crucial in medical diagnosis scenarios, where it's essential to present only the most relevant diagnostic features from input (e.g., an X-ray) and thereby avoiding cognitive overload for human experts.

S3. The method leverages a surrogate model to estimate the quality of downstream decisions, providing a practical and scalable solution when large-scale human supervision is either expensive or impractical. SLOG further uses this surrogate-estimated quality to enhance its guidance generation by introducing an augmented loss term into the learning objective, effectively refining outputs from the vision-language model (VLM).

S4. The authors demonstrate the effectiveness of their approach on a real-world medical diagnosis dataset with significant improvements in downstream decision quality (especially measured as F1 scores across different pathologies and in overall performance). For medical diagnosis applications, recall is highly critical and it shows improvement for the proposed method.

**Weaknesses**

W1. The effectiveness of SLOG is heavily dependent on the quality of the surrogate model. If the annotated data used to train the surrogate model is biased or insufficient, the model may fail to generalize effectively. Any inaccuracies or biases in estimating decision quality can misguide the learning process for generating actionable guidance from the VLM, potentially leading to system collapse. The method relies on the assumption that the surrogate model imitates the full complexity of human reasoning which may not always hold true.

W2. While the authors train the surrogate model using only 10% of the dataset, this amount is relatively large compared to many high-stakes tasks where practically such scale data may not even be available. The authors do not conduct ablation experiments to evaluate the surrogate model’s performance at varying data scales. This leaves unanswered questions about how the amount of annotated training data influences both the surrogate model's effectiveness and the quality of the generated guidance.

W3. The experiments in the study are currently limited to using a single model (R2Gen) and a single dataset (MIMIC-CXR-IV) on the medical diagnosis task. The current findings show that the method is promising, however, I think there is a need to extend the experiments to a few more tasks and models to assess the generalizability and robustness of the proposed method.

W4. Following up on the previous point: There are many pretrained VLMs specifically adapted to the biomedical domain (e.g. LLaVa-Med v1.5). I think experiments on directly using pretrained VLMs would be more relevant for the community broadly as many practical applications directly involve fine-tuning pretrained models for downstream tasks.

W5. The authors use the BLEU metric to measure the informativeness of guidance. However, BLEU has been widely criticized in translation literature for its inability to effectively capture semantic similarity. This may explain why the BLEU scores across different configurations appear similar. The authors should also report additional evaluation metrics including model-based semantic metrics and human evaluation on a subset to assess the informativeness of the guidance. The proposed method doesn't perform poorly on 5 classes out of 14 and more analysis should be done to understand the reason for failure cases.

W6. Although the quantitative results suggest that the method is effective, a more in-depth analysis of the guidance generated is necessary. It would be valuable to identify if there are specific characteristics noticeable across different configurations. Furthermore, it is worth investigating whether specific tokens disproportionately influence the surrogate model's quality estimation. If such tokens exist, the method might be vulnerable to anchor bias, thereby undermining the robustness of the guidance generation process.

---

> ### Author Response · Authors · 2025-04-26
>
> Thank you for taking the time to evaluate our contribution.  We really appreciate your valuable comment and below we address the concerns that you have raised.
>
> **Single model and dataset.**
>
> We agree that reporting the performance of slog based on more than one model seems more convincing. Hence we also finetuned another model R2GenCMN and we report the results on  Table 3, Table 4, and Figure 4.
>
> **Ablation studies.**
>
> We appreciate the insightful feedback put forward by the reviewer.  The ablation study was indeed important and updated the paper with ablation study. In particular, we conducted the ablation with varying size of data used to train the surrogate ($1\%$, $5\%$, $10\%$, ) and analyze the performance of the finetuned model on the test set. We reported our results in the Table 7, Figure 5 and appendix A.
>
> **BLEU Score.**
>
> We agree that BLEU score has several limitations and therefore we added another model based evaluation metric BLEURT. We updated Table 4 with the inclusion of BLEURT.
>
> **Human Evaluation.**
>
> We absolutely agree that human evaluation would be remarkably beneficial, however, that said, as the reviewer also signalled, involving humans is a time-consuming and costly job, we couldn’t manage to conduct human evaluation in this short period.
>
> **Using Pretrained VLM.**
>
> Using pre-trained vlms under the light of slog is an interesting direction. However, it was infeasible to carry additional experiments out within the two week timeframe allowed for the rebuttal, especially considering we had to implement additional experiments also requested by the other reviewers as well.

---

### Author Response · Authors · 2025-04-26

We thank all the reviewers for their effort and valuable feedbacks. As per the reviwers' concerns over multiple issues, we have made certain modifications and inclusions to our paper. All the changes are marked with red color.

---

### Decision · Action_Editor_fXAu · 2025-07-10

**Recommendation:** Reject

**Additional Comments:**

Overall, the paper and project is perceived to be interesting and relevant, and I would like to encourage the authors to submit a revision. Reviewers and AE feel that the paper is close to the acceptance threshold and with a few additions could be a clear accept. In particular, I recommend:
- [strongly recommended] human evaluation (could be a sample, could be performed by paper authors)
- [recommended] authors are encouraged to explore using a pretrained VLMs (LAVA, as per reviewer 3fxa's suggestion)
- [encouraged] reviewers feel that an additional dataset could strengthen the contributions substantially; AE feels it would be beneficial for arguing for acceptance, but not in their view strictly necessary.

**Audience:**

Yes

**Audience Explanation:**

Reviewers feel the work would be interesting to readers focusing on the medical domain, and several feel it meets the criterion for acceptance; however AE feels there is opportunity to strengthen the work further.

**Claims And Evidence:**

No

**Claims Explanation:**

By and large, reviewers felt that the work has merit and interest, although the scope is somewhat limited to the medical domain. Reviewers appreciated the discussion with authors, although they also felt some of their questions were not addressed, and some clarity on what will be included in the next draft was lacking. I too feel some lack of clarity regarding what exactly will be included in the next version of the work.

After the discussion phase with reviewers, several questions were raised about the strength of the evaluation provided, with two reviewers requesting human evaluation (3fxa, NG23). Both reviewers felt that a performing human evaluation on a sample of data would be sufficient to motivate acceptance, but this has not been contributed. The authors are thus encouraged to run a small human annotation sample (potentially even amongst themselves), for the next version of the submission.

Reviewer 3fxa also recommended the inclusion of an additional model baseline, which I agree would strengthen the contribution.

**Resubmission Of Major Revision:**

The authors may consider submitting a major revision at a later time.